# Weather in the Hungarian Lowland from the Point of View of Humans

**Ferenc Ács [1],\*, Erzsébet Kristóf [2], Annamária Zsákai [3], Bertold Kelemen [1], Zita Szabó [1] and Lara Amanda Marques Vieira [4]**

[1] Department of Meteorology, Faculty of Science, Institute of Geography and Earth Sciences, Eötvös Loránd University, 1117 Budapest, Hungary; kelemen.berti@gmail.com (B.K.); szabo.zita1999@gmail.com (Z.S.)

[2] Excellence Center, Faculty of Science, Eötvös Loránd University, 2462 Martonvásár, Hungary; ekristof86@caesar.elte.hu

[3] Department of Human Anthropology, Faculty of Science, Eötvös Loránd University, 1117 Budapest, Hungary; zsakaia@caesar.elte.hu

[4] Department of Meteorology, Institute of Astronomy, Geophysics and Atmospheric Sciences, University Sao Paulo, São Paulo 05508-090, Brazil; laramanda96@gmail.com

\* Correspondence: acs@caesar.elte.hu; Tel.: +3-630-292-9408

**Abstract:** Weather at different locations in the Hungarian lowland in different seasons (winter, summer) and times of day (morning, noon) is investigated from the human biometeorological point of view. Human thermal load characteristics of weather are described in terms of clothing resistance and operative temperature. Individual human thermal load–thermal sensation relationships have been estimated to study weather variation in the cities of Sopron (cooler part of Hungary) and Szeged (warmer part of Hungary). In the clothing resistance model, the humans are walking at a speed of 1.1 ms$^{-1}$ in outdoor conditions without sweating. The main findings are as follows. (1) In the early summer mornings, the weather is sensed as "neutral" or "cool", in these cases the inter-person variation effect is very small. (2) At noon in summer, heat stresses (clothing resistance parameter values less than $-2$ clo) are registered. In these cases, high temperature and irradiation, as well as low or moderate wind, characterized the atmospheric environment. Then, the inter-person variation effect is clearly visible. (3) The strength of summer heat excess at noon seems to be larger than the strength of winter heat deficit in the early morning. (4) Clothing resistance differences caused by inter-person variations and by weather variations between the cities of Sopron and Szeged are comparable in the majority of cases. When they are not comparable, the site variation effect is much larger than the inter-person variation effect. The clothing resistance model is constructed for individual use and it can be equally applied on both weather and climate data.

**Keywords:** clothing resistance; operative temperature; thermal load; thermal sensation; metabolic heat flux density; Hungarian lowland; weather

## 1. Introduction

Today, there are dozens of human thermal indices [1,2]. Initially, they were represented mostly as single parameters (e.g., air temperature [3], dew point temperature [4], apparent temperature [5]), or by their combination (e.g., discomfort index [6], effective temperature [7], air enthalpy [8]). Later on, however, the human body energy balance-based methods became increasingly popular [8–13]. In most of these approaches not only environmental thermal load [14–18] but also human thermal sensation [19–27] is characterized. Among bioclimatic indices, Predicted Mean Vote (PMV), Physiological Equivalent Temperature (PET) and Universal Thermal Climate Index (UTCI) are the most popular. The same indices are most frequently used in Hungary also [28,29]. They were used to characterize the topic of human thermal comfort. For instance, the following are extensively discussed: (a) the effect of variations of microclimate caused by the complex

urban environment [30–32], (b) the effect of deviation between different urban districts in Budapest [33], (c) the effect of thermal contrast between urban and rural areas that lie close to the city [34,35], (d) the topic of subjective thermal sensation [36], (e) the phenomenon of heat stress during heatwave periods [37], and (f) the effect of climate change on the human thermal environment in some chosen Hungarian cities [29]. Lastly, an application of PET for calculating Tourism Climatic Index is also outlined [38].

The humans considered are "standardized". So, for instance, the human in the most widespread PET index is a man of 35 years old with a body weight of 75 kg and a body length of 175 cm. This human is very similar to the UTCI-Fiala model human [39] used in the UTCI index. Since the "standardized" human is considered, inter-person variation effects are not treated at all. Specific persons have been considered only recently in the works of Ács et al. [40,41]. In these analyses, the strength of the thermal load is considered in terms of operative temperature and clothing resistance, treating humans in natural outdoor conditions where they are walking without sweating. Personal state variables are taken from a Hungarian human dataset [42]. The persons are also characterized from the point of view of their body shape [41]. The somatotypes are classified using the Heath–Carter somatotype classification method. The analyses refer either to the location [40] or to the Carpathian region [41].

In the works of Ács et al. [18,19], thermal sensations related to the thermal loads are not treated at all. To bridge this gap, the aims of this study are as follows: (a) to construct individual human thermal load–thermal sensation relationships by concurrent collection of weather and thermal sensation data, (b) to analyze weather events from the point of view of individual human thermal loads and sensations, (c) to compare the effect of inter-person variations and the effect of weather variations between two thermally opposite locations in Hungary on the evolution of individual human thermal loads and sensations. The cities of Sopron and Szeged are chosen to represent the two thermally opposite locations. The methods used are described in Section 2. The locations where weather and thermal sensation data are collected are briefly introduced in Section 3. Basic information regarding weather and human data can be found in Section 4. The results are presented and analyzed in Section 5. The topic and the main findings are discussed in Section 6. The conclusions are drawn in Section 7.

## 2. Methods

Three topics will be briefly presented: the clothing resistance model, the Heath–Carter somatotype classification method and the basic considerations in the management of the data referring to the relationship between thermal sensation and thermal load.

### 2.1. Clothing Resistance Model

The physics of the clothing resistance model are discussed in detail in the works of Ács et al. [40,43]. The scheme is based on clothed human body energy balance considerations treating the human body as simply as possible with a one-node model [44]. Only the clothed human body–air environment exchange processes are treated; there is no treatment of the thermoregulatory system at all. The human body (37 °C) and skin (34 °C) temperatures are used as boundary conditions. The scheme supposes that (a) the clothing completely covers the human body, (b) it adheres strongly to the skin surface, and that (c) the albedo of clothing is equal to the albedo of skin. Non-sweating, walking humans at a speed of 1.1 m·s$^{-1}$ (4 km·h$^{-1}$) in outdoor conditions are treated. Here, only the basic equations for calculating clothing resistance ($r_{cl}$), operative temperature ($T_o$), radiation energy balance and metabolic heat flux density of walking humans ($M$) are presented,

$$r_{cl} = \rho \cdot c_p \cdot \frac{T_S - T_o}{M - \lambda E_{sd} - \lambda E_r - W} - r_{Hr}, \tag{1}$$

$$T_o = T_a + \frac{R_{ni}}{\rho \cdot c_p} \cdot r_{Hr}, \tag{2}$$

where $\rho$ is air density (kgm$^{-3}$), $c_p$ is specific heat at constant pressure (Jkg$^{-1}$ °C$^{-1}$), $T_S$ is skin temperature (°C) (a constant, 34 °C), $r_{Hr}$ is the combined resistance for expressing thermal radiative and convective heat exchanges (sm$^{-1}$), $T_a$ is air temperature (°C), $R_{ni}$ is the isothermal net radiation flux density (Wm$^{-2}$), $\lambda E_{sd}$ is the latent heat flux density of dry skin (Wm$^{-2}$), $\lambda E_r$ is the respiratory latent heat flux density (Wm$^{-2}$) and $W$ is the mechanical work flux density (Wm$^{-2}$) referring to the activity under consideration, in this case walking.

Net radiation is estimated by the isothermal net radiation approach as

$$R_{ni} = S \cdot (1 - \alpha_{cl}) + \epsilon_a \sigma T_a^4 - \epsilon_{cl} \sigma T_a^4, \tag{3}$$

where $S$ is global radiation, $\alpha_{cl}$ is clothing albedo, $\epsilon_a$ is atmospheric emissivity and $\epsilon_{cl}$ is the emissivity of clothing or skin and $\sigma$ is the Stefan–Boltzmann constant. In our scheme, $\alpha_{cl} = 0.25$–$0.27$, $\epsilon_{cl} = 1$. Global radiation is estimated via relative sunshine duration *rsd* according to Mihailović and Ács [45].

$$S = Q_0 \cdot [\alpha + (1 - \alpha) \cdot rsd], \tag{4}$$

where $Q_0$ is the global radiation constant (MJ·m$^{-2}$·hour$^{-1}$) referring to clear sky conditions and a 1-h time period and $\alpha$ is the corresponding dimensionless constant referring to the same hour.

The combined resistance $r_{Hr}$ of radiative $r_R$ and convective $r_{Ha}$ heat exchanges is given by

$$\frac{1}{r_{Hr}} = \frac{1}{r_{Ha}} + \frac{1}{r_R}, \tag{5}$$

$$r_{Ha}\left[\text{sm}^{-1}\right] = 7.4 \cdot 41 \sqrt{\frac{D}{U_{1.5}}}, \tag{6}$$

$$\frac{1}{r_R} = \frac{4\epsilon_{cl}\sigma T_a^3}{\rho c_p}, \tag{7}$$

where $D$ (m) is the diameter of the cylindrical body used to approximate the human body, $U_{1.5}$ is air speed relative to the human body at 1.5 m (around chest height). $U_{1.5}$ is calculated from the wind speed at a height of 10 m.

According to Weyand et al. [46], a walking human's $M$ in (W) can be calculated as,

$$M = M_b + M_w, \tag{8}$$

where $M_b$ is the basal metabolic rate (sleeping human) and $M_w$ is the metabolic rate referring to walking. Both terms can be estimated if sex, age (year), body mass $M_{bo}$ (kg) and body length $L_{bo}$ (cm) of the human considered are known. Different parameterizations for $M_b$ were thoroughly reviewed in the work of Frankenfield et al. [47], where it is suggested that the parameterization of Mifflin et al. [48] is one of the best:

$$M_b^{male}\left[\text{kcal·day}^{-1}\right] = 9.99 \cdot M_{bo} + 6.25 \cdot L_{bo} - 4.92 \cdot age + 5, \tag{9}$$

$$M_b^{female}\left[\text{kcal·day}^{-1}\right] = 9.99 \cdot M_{bo} + 6.25 \cdot L_{bo} - 4.92 \cdot age - 161. \tag{10}$$

To get $M_b$ in (Wm$^{-2}$), the human body surface $A$ (m$^2$) also has to be estimated. The parameterization of Dubois and Dubois [49] is used for estimating $A$ (m$^2$) taking $M_{bo}$ and $L_{bo}$ as inputs,

$$A = 0.2 \cdot M_{bo}^{0.425} \cdot \left(\frac{L_{bo}}{100}\right)^{0.725}. \tag{11}$$

$M_w$ is parameterized according to Weyand et al. [46] as follows,

$$M_w = 1.1 \cdot \frac{3.80 \cdot M_{bo} \cdot \left(\frac{L_{bo}}{100}\right)^{-0.95}}{A}. \tag{12}$$

Formula (1) in the work of Weyand et al. [46] refers to a walking distance of 1 m. Since the reference walking speed in our model is 1.1 ms$^{-1}$, formula (1) in [46] is multiplied by the factor 1.1, and by dividing this by $A$, we get $M_w$ in (Wm$^{-2}$). The $\lambda E_{sd} + \lambda E_r$ sum can be expressed as a function of M according to Campbell and Norman [50]. $W$ is parameterized according to Auliciems and Kalma [51].

### 2.2. Heath–Carter Somatotype Classification Method

The Heath–Carter somatotype characterizes the human morphological body shape by using three biologically interpretable components. The endomorphy (1st) component describes the relative fatness of the whole body, the mesomorphy (2nd) component estimates the skeleton–muscular robusticity, while the ectomorphy (3rd) component evaluates the linearity of the human body [52]. The somatotype components of an individual can be estimated by using anthropometric dimensions. The natural combinations of the three components help us to describe the variability of the human morphological body shape in a three-dimensional coordinate system of the three continuous scales of the components (traditionally, values around 4 represent the normal development of the components). The main advantage of somatotyping is that the somatotype of individuals who significantly differ in their absolute body dimensions may have very similar somatotypes, so the method can characterize size-independent body morphology. To facilitate understanding of the variability of body shape, a two-dimensional somatochart is used to visualize the individual somatoplots (the x and y coordinates of the somatotypes are derived from the 3 somatotype components). By considering the dominant relationships of the components, Carter introduced 13 somatotype categories [53]. The names of the categories usually indicate the dominant components (e.g., balanced endomorph, mesomorph–endomorph somatotypes), sometimes the dominant relationship of the smaller components is used as the attributive in the category name (e.g., mesomorphic endomorph somatotype).

### 2.3. Thermal Load and Thermal Sensation Data Treatment

Observations of each individual were divided into thermal sensation type groups: very cold, cold, cool, neutral, slightly warm, warm and very warm. Some observations were considered as outliers. Data were filtered as follows: first, observations were removed if negative (positive) clothing resistance values were associated with thermal sensation types very cold, cold and cool (slightly warm, warm and very warm), second, observations of each thermal sensation group below the 5th and above the 95th percentiles of the associated $T_o$ values were considered as outliers—therefore, those were omitted from the analysis.

## 3. Locations

Concurrent collection of weather data and subjective thermal sensations is performed in the cities of Budapest ($\varphi = 47°29'$; $\lambda = 19°9.5'$), Gödöllő ($\varphi = 47°36'$; $\lambda = 19°21'$), Martonvásár ($\varphi = 47°19'$; $\lambda = 18°47.5'$) and Hajdúböszörmény ($\varphi = 47°40'$; $\lambda = 21°31'$). Weather observed in cities of Sopron ($\varphi = 47°41'$; $\lambda = 16°35'$) and Szeged ($\varphi = 46°15'$; $\lambda = 20°10'$) is characterized in terms of clothing resistance and individual human thermal sensation. The locations of the cities are presented in Figure 1.

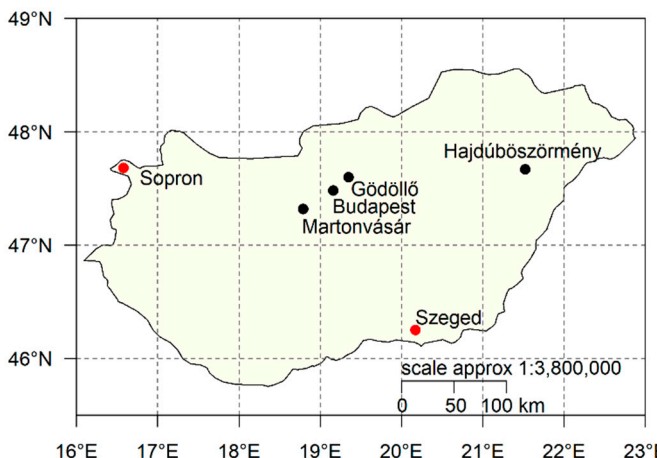

**Figure 1.** Locations of the cities where the collection of weather data is carried out and the thermal sensation observations are performed.

Data collection is performed by the authors of this study. The cities chosen represent the places of residence of the authors, except the cities of Sopron and Szeged. The cities of Sopron and Szeged are chosen because of their thermal contrast. Sopron belongs among the coldest, whilst Szeged among the warmest, cities in the country.

## 4. Data

### 4.1. Weather Data

Weather data relevant from the point of view of thermal load are as follows: air temperature, global radiation or relative sunshine duration, cloudiness, air humidity, wind speed and atmospheric pressure. They are taken from the nearest automatic meteorological station operated either by the Hungarian Meteorological Service (HMS) or by a private company "Időkép". The beeline distance between the automatic stations and the thermal sensation observation locations was in all cases shorter than 3 km. Special care is taken to avoid microclimatic influences on the thermal sensation observation locations, so the representative nature of the meteorological data at the thermal sensation observation location is carefully checked in all cities and at all times. Weather data collected for the cities of Sopron and Szeged refer to two 31-day long winter and summer season periods at noon (12 UTC) (13 December 2018–12 January 2019; 12 June 2019–12 July 2019) and early in the morning (6 UTC) (14 January 2020–14 February 2020; 14 July 2019–13 August 2019).

### 4.2. Human Data

Three types of human data are distinguished: (1) human state variables (sex, age, body mass and body length) needed for calculating resting, walking and total metabolic heat flux densities (Table 1), (2) data needed for estimating body shape type components, and (3) thermal sensation data. Data types 1 and 2 are collected for six humans in total, two males and four females. Data collection is made in the Laboratory of the Department of Biological Anthropology of Eötvös Loránd University. Subjective thermal sensation estimation is performed in that 10 min period during which meteorological data are taken from the HMS or "Időkép" website. The observers took care of their thermal history, the clothes worn were appropriate for the weather, and the subjective estimation took place during walking. A seven-grade scale is used for the estimation, and the grades are marked as "very cold", "cold", "cool", "comfortable", "slightly warm", "warm" and "very warm".

**Table 1.** Human state variables together with $M_b$ (basal), $M_w$ (walking), $M$ (total, $M_b + M_w$), humans: Person 1 (P1), Person 2 (P2), Person 3 (P3), Person 4 (P4), Person 5 (P5) and Person 6 (P6).

| Persons | Sex | Age (Years) | Body Mass (kg) | Body Length (cm) | Basal Metabolic Heat Flux Density (Wm$^{-2}$) | Walking Energy Flux Density (Wm$^{-2}$) | Total Energy Flux Density (Wm$^{-2}$) |
|---|---|---|---|---|---|---|---|
| Person 1 | male | 64 | 89.0 | 190.0 | 40.8 | 94.5 | 135.3 |
| Person 2 | female | 34 | 64.5 | 160.5 | 38.6 | 103.9 | 142.5 |
| Person 3 | female | 45 | 68.7 | 165.1 | 37.3 | 102.7 | 140.0 |
| Person 4 | male | 24 | 80.0 | 176.0 | 44.6 | 100.7 | 145.3 |
| Person 5 | female | 20 | 55.0 | 169.0 | 40.6 | 86.9 | 127.5 |
| Person 6 | female | 24 | 70.6 | 173.8 | 40.0 | 94.0 | 134.0 |

## 5. Results

The results characterize human body shape types for each human separately, the individual human thermal load–thermal sensation relationships and the variation of weather in the cities of Sopron and Szeged in terms of individual human thermal load and sensation. As it is mentioned, the thermal climate differences between the cities of Sopron and Szeged are among the largest in the Hungarian lowland and this is why they are chosen for the analysis.

### 5.1. Heath–Carter Somatotype Classification Results

Person 2 and Person 4 have a typical endomorph body shape with increased skeleton–muscular robusticity. Person 1 and Person 3 have a mesomorph–endomorph body shape; this type of somatotype category is characterized by the dominance of both endo- and mesomorphy components, namely, increased body fatness and increased skeleton–muscular robusticity. However, Person 1′s somatotype reveals less extremity in endomorphy and mesomorphy; his somatotype is very close to the central somatotype, in which the three components form the body shape equally (Figure 2, Table 2). Person 5′s somatotype is ectomorph–endomorph; in her case, the normal limb–trunk ratio and the normal level of body fatness exist with decreased skeleton–muscular development. Person 6′s somatotype is also endomorph, but in this case, it is a balanced endomorph, when relative fatness is dominant in the body shape and linearity and skeleton–muscular robusticity are equally underdeveloped.

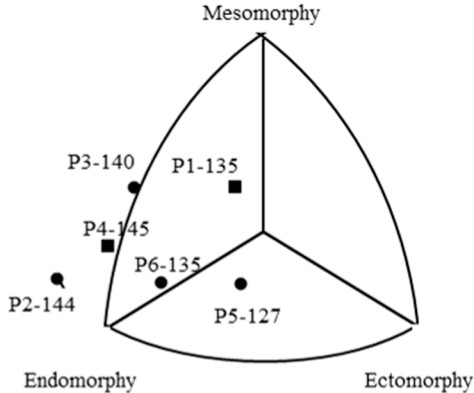

**Figure 2.** The individual somatoplots of the studied persons (P1–6, the numbers after the hyphens indicate the individual M values in Wm$^{-2}$, ■: male subjects, ●: female subjects).

**Table 2.** The studied person's somatotype characteristics in the order of metabolic activity (M).

| Persons | Sex | Somatotype Components | Somatotype Category | M (Wm$^{-2}$) |
|---|---|---|---|---|
| P5 | female | 4.5–2.5–4.0 | ectomorph–endo-morph | 127.5 |
| P1 | male | 3.5–4.0–2.0 | mesomorph–endomorph | 135.3 |
| P3 | female | 6.0–5.0–1.0 | mesomorph–endomorph | 140.0 |
| P2 | female | 9.0–3.5–1.0 | mesomorph–endomorph | 144 |
| P4 | male | 7.5–4.0–1.5 | mesomorph–endomorph | 145.3 |
| P6 | female | 6.1–2.4–2.0 | balanced-endomorph | 134.0 |

By considering the individual M values of the studied persons, as a tendency it can be stated that M increases with decreasing ectomorphy; the higher the level of endomorphy, or the higher the level of mesomorphy, the higher the value of M.

*5.2. Individual Thermal Sensation–Thermal Load Relationships*

Among the six humans considered, those two humans were chosen for whom the M difference is the largest. Accordingly, humans 4 and 5 were chosen. The individual $r_{cl}$–$T_o$–thermal sensation scatter-point chart is presented in Figure 3a. Thermal sensation types are denoted by color; humans are distinguished by symbol (human 4 square, human 5 circle). Examining Figure 3a carefully, we can see that the point-cloud of human 4 is somewhat above (larger $r_{cl}$ values) the point-cloud of human 5 in a warm zone (operative temperature above 40 °C) and somewhat below (smaller $r_{cl}$ values) in the cold zone (operative temperature below 10 °C). This results from the difference between their M values. In the zone of 10 °C < $T_o$ < 40 °C, this systematic deviation decreases and vanishes in the zone of neutral thermal sensation (20 °C < $T_o$ < 30 °C). Let us now see the overlap between the different thermal sensation types for one person (different colors, the same symbol) and between two persons (different colors, different symbols). In the cold zone, both humans 4 and 5 sensed the environment's thermal load as "cool" and "cold", but for the most part "cool". In the warm zone, human 4 sensed the environment's thermal load as "slightly warm", "warm" and "very warm", but "very warm" only in two cases. Human 5 sensed similar conditions mostly as "warm" and "very warm" and less frequently as "slightly warm". It may be observed that human 5 perceived "very warm" many more times than human 4. In the zone of 10 °C < $T_o$ < 40 °C, it is hard to separate the "neutral" and "cool" thermal sensation types with a fixed boundary value in the region of positive $r_{cl}$ values, and, similarly, "neutral" and "slightly warm" thermal sensation types in the region of negative $r_{cl}$ values. It seems to be that much more observations are needed for such separations, if they are at all possible.

Thermal sensation–thermal load dependence can also be analyzed by presenting the thermal sensation–$r_{cl}$ point-cloud; this is given in Figure 3b. The points denoted by squares and circles refer to human 4 and 5, respectively. As we expected, the overlap between the $r_{cl}$ values of human 4 and 5 is good for the thermal sensation categories "neutral", "cool" and "slightly warm". The same overlap is much weaker for the thermal sensation categories "very warm", "warm" and "cold". There are also transitional $r_{cl}$ values, when adjacent thermal categories can equally happen. For instance, for human 5, thermal categories "neutral" and "cool" for 0.2 clo < $r_{cl}$ < 0.5 clo, thermal categories "cool" and "cold" for 0.6 clo < $r_{cl}$ < 1.2 clo, thermal categories "slightly warm" and "warm" for −1.2 clo < $r_{cl}$ < −0.5 clo and thermal categories "very warm" and "warm" for −3.5 clo < $r_{cl}$ < −2.0 clo.

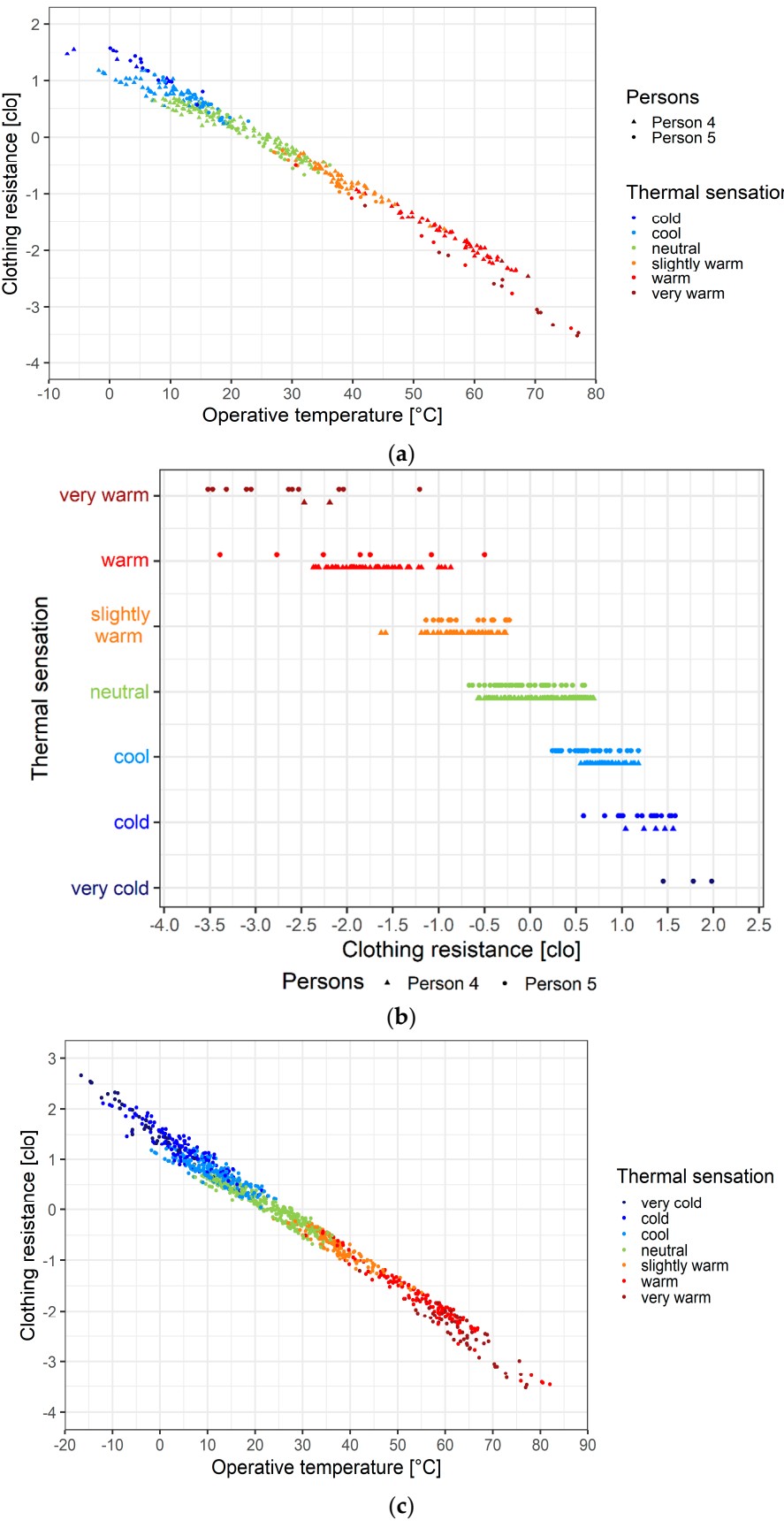

**Figure 3.** *Cont.*

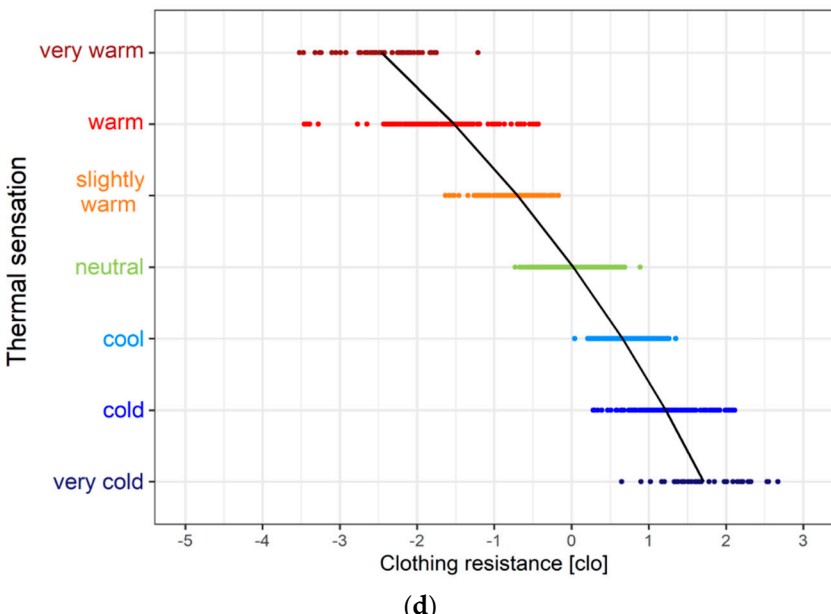

(**d**)

**Figure 3.** (**a**) The $r_{cl}$–$T_o$–thermal sensation relationships for persons 4 and 5; (**b**) The thermal sensation-$r_{cl}$ relationships for persons 4 and 5; (**c**) The $r_{cl}$–$T_o$–thermal sensation relationships for all Hungarian persons; (**d**) Dependence between thermal sensation and $r_{cl}$ for all persons considered.

The $r_{cl}$–$T_o$–thermal sensation relationships for all Hungarian persons are presented in Figure 3c.

Data referring to the Brazilian person are not included in Figure 3c,d; they differ considerably from the Hungarian data, therefore, a larger dataset needs to be constructed to explain the deviations. The $r_{cl}$–$T_o$ relationship is similar to a band. The formation of the band is caused by the vertical and horizontal spread of the points. The vertical spread of the points is caused by variations of M, whilst the horizontal spread by variations of weather. The vertical bandwidth is about 0.5 clo, which is caused by the variations of $M$ (between 127 and 145 $\text{Wm}^{-2}$) of the five persons. Each thermal sensation type can be characterized in terms of $r_{cl}$ and $T_o$, such a subjective estimate is given in Table 3.

**Table 3.** Operative temperature and clothing resistance limits for different thermal sensation types obtained for 5 Hungarian persons.

| Thermal Sensation Type | Operative Temperature ($T_o$) Range (°C) | Clothing Resistance ($r_{cl}$) Range (clo) |
|---|---|---|
| very warm | $T_o > 65$ | $r_{cl} \leq -2.4$ |
| warm | $45 < T_o < 60$ | $-2 < r_{cl} < -1.1$ |
| slightly warm | $30 < T_o \leq 45$ | $-1.1 \leq r_{cl} < -0.5$ |
| neutral | $15 < T_o \leq 30$ | $-0.5 \leq r_{cl} \leq 0.5$ |
| cool | $5 \leq T_o \leq 15$ | $0.5 < r_{cl} \leq 1.2$ |
| cold | $-8 < T_o < 5$ | $1.2 < r_{cl} < 1.7$ |
| very cold | $-8 \geq T_o$ | $1.7 \leq r_{cl}$ |

There are also transitional zones; for instance, between the thermal sensation types "very warm" and "warm", which makes sense, since human thermal sensation is not discrete. In such cases, both adjacent thermal sensation types can equally be used. It is also noticeable that more observations are needed for the thermal sensation type "very cold".

The thermal sensation–$r_{cl}$ dependence for all Hungarian persons considered is presented in Figure 3d.

According to the curve plotted, the typical thermal sensation type–$r_{cl}$ correspondence is as follows: for "very warm" $r_{cl} \leq -2.4$ clo, for "warm" $r_{cl} = -1.5$ clo, for "slightly warm" $r_{cl} = -0.8$ clo, for "neutral" $r_{cl} = 0$ clo, for "cool" $r_{cl} = 0.7$ clo, for "cold" $r_{cl} = 1.2$ clo and

for "very cold" $r_{cl} \geq 1.7$ clo. Lastly, the user can find all these data in a database in the Supplementary Material.

*5.3. Weather Variations from the Point of View of Individual Human Thermal Loads and Sensations*

The variation of weather in 31-day winter and summer season periods separately at noon and early in the morning for the city of Sopron in terms of clothing resistance is presented in Figure 4a,b, respectively. As mentioned, thermal load is represented for humans 4 and 5. In the Hungarian lowland, typical winter mean clothing resistance values of an "average Hungarian female" are between 1.4 and 1.6 clo [41]. The fluctuations registered in Sopron in the winter period at noon (Figure 4(a1)) in most of the cases are very close to these values. This is mostly sensed by humans 4 and 5 as "cold", only in some cases as "cool". Nevertheless, there are also warmer cases (days 14, 15, 16 and 27), when the $r_{cl}$ values are between +0.5 and −0.5 clo. At noon on 8 January 2019 (case 27), the outdoor environment's operative temperature (25.7 °C) was much higher than the air temperature (air temperature 0 °C) because of greater global radiation (373 Wm$^{-2}$) and gentle wind (0.8 ms$^{-1}$). This is sensed by both humans as "neutral". Typical summer mean clothing resistance values in the Hungarian lowland are around 0 clo [43]; consequently, the corresponding thermal sensation is "neutral". However, at noon in summer, the thermal load can be much larger depending on radiation and wind speed conditions. This can be seen in the example of the city of Sopron (Figure 4(a2)). $r_{cl}$ fluctuates between −0.5 and −2.5 clo, but in the majority of the cases $r_{cl}$ is lower than −1 clo. The corresponding thermal sensations vary from "neutral" ($r_{cl}$ values above −0.5 clo) through "slightly warm" ($r_{cl}$ values around −1 clo) to "warm" ($r_{cl}$ values around −1.5 clo) or "very warm" ($r_{cl}$ values towards −2.5 clo). The strongest thermal load was on the 10th day (21 June 2019), the weakest thermal load was on the 31st day (13 July 2019).

The main difference between them is in their radiation and wind speed conditions. When the thermal load is strong ($r_{cl}$ is −2.5 clo or lower), global radiation is very high (about 800 Wm$^{-2}$) and wind speed is smaller (around 1 ms$^{-1}$). In the case of smaller thermal loads ($r_{cl}$ is between −0.5 and −1 clo), global radiation is much lower (250–300 Wm$^{-2}$) and wind speed is moderate (around 3 ms$^{-1}$) or even greater. Note that in such situations, global radiation and wind speed fluctuations can be high; consequently, the thermal load and sensation changes can also be significant. This is well illustrated in Table 4, where the thermal load and sensation data refer to 10 August 2020 at noon (12 UTC).

Four cases are distinguished in this observation. They differ only in terms of solar exposure and/or wind speed values. In the shade *rsd* = 0, in the sun *rsd* = 1, the average wind speed is 1.7 ms$^{-1}$, the wind gust is 3.1 ms$^{-1}$. For average wind speed in the shade (global radiation is 283.6 Wm$^{-2}$) (case 1), the simulated $T_o$ and $r_{cl}$ values are 42.8 °C and −1.01 clo, respectively. Then person P1 reported the thermal sensation "slightly warm". For the same wind, but in the sun (global radiation is 766.4 Wm$^{-2}$) (case 2) $T_o$ is 69.1 °C and $r_{cl}$ is −2.60 clo. In this case, the reported thermal sensation was "very warm". In the case of a wind gust, the simulated thermal loads were smaller. In the shade (case 3), $T_o$ is 41.3 °C, which is 1.5 °C lower than in case 1, and $r_{cl}$ is −0.85 clo, which can be even sensed as "neutral" for that short moment. In the sun (case 4), $T_o$ is 63.6 °C, which is 5.5 °C lower than in case 2 and the related $r_{cl}$ value is −2.20 clo. As we can see, the cooling effect of a wind gust in sunny situations can be significant. Nevertheless, the effect of wind variation on thermal load is much less than the effect of global radiation variation. This is unequivocally visible when comparing the $r_{cl}$ values in case 1 and case 2 (radiation effect for lower wind speed value) and the $r_{cl}$ values in case 2 and case 4 (wind effect in sunny conditions).

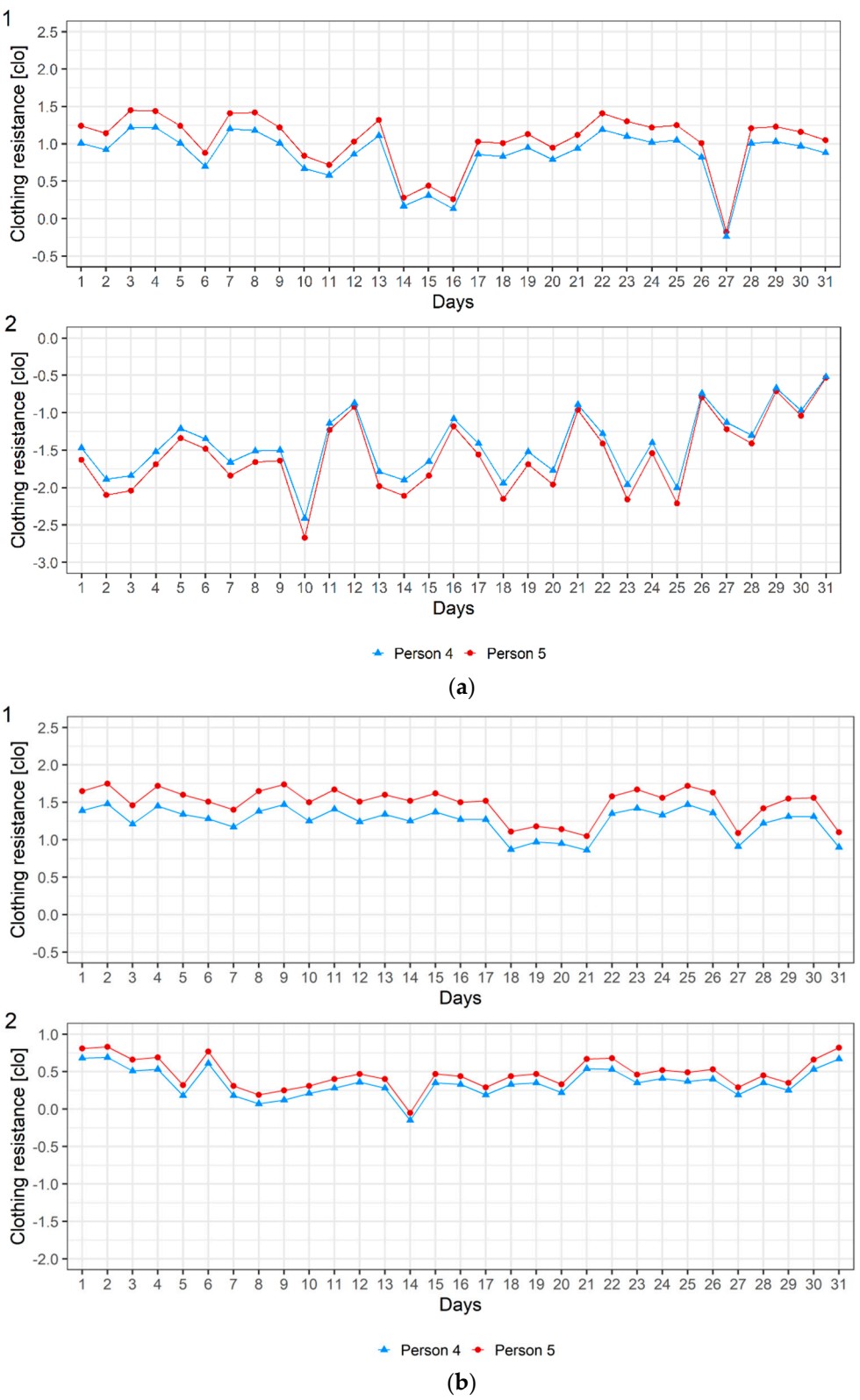

**Figure 4.** (**a**) Evolution of the thermal load at noon (12 UTC) for humans 4 and 5 over 31-day winter (**1**) and summer (**2**) season periods in the city of Sopron; (**b**) Evolution of the thermal load in the early morning (4–6 UTC) for humans 4 and 5 over 31-day winter (**1**) and summer (**2**) season periods in the city of Sopron.

**Table 4.** Meteorological conditions, thermal load and thermal sensation characteristics in Martonvásár on 10 August 2020 at noon (12 UTC). Human data refer to person P1. Symbols: $T_a$—air temperature, $rsd$—relative sunshine duration, $N$—cloudiness, $rh$—relative humidity of air, $p$—atmospheric pressure, $T_o$—operative temperature, $r_{cl}$—clothing resistance.

| Case | Meteorological Conditions | | | | | | Thermal Load | | Thermal Sensation |
|------|------|------|------|------|------|------|------|------|------|
| | $Ta$ (°C) | $rsd$ | $N$ | Wind (ms$^{-1}$) | $rh$ (%) | $p$ (hPa) | $T_o$ (°C) | $r_{cl}$ (clo) | |
| 1 | 33.1 | 0 | 0.4 | 1.7 | 44 | 1015 | 42.8 | −1.01 | slightly warm |
| 2 | 33.1 | 1 | 0.4 | 1.7 | 44 | 1015 | 69.1 | −2.60 | very warm |
| 3 | 33.1 | 0 | 0.4 | 3.1 | 44 | 1015 | 41.3 | −0.85 | |
| 4 | 33.1 | 1 | 0.4 | 3.1 | 44 | 1015 | 63.6 | −2.20 | |

The early morning results in the winter and summer season periods over the course of 31 days are presented in Figure 4b. Note that the early morning results in the winter period refer to the year 2020, that is, the results relating to early morning and noon do not refer to the same day. In the great majority of the cases, $r_{cl}$ is around 1.5 clo, the corresponding thermal sensation is "cold", and only rarely "cool". The lowest $r_{cl}$ values are around 1 clo, which are sensed (almost) always as "cool" by humans 4 and 5. These cases are caused by milder air temperatures (temperatures between 7 and 11 °C). In summer, in the early morning, there is no heat load, or it is very weak. This environmental load is sensed by both humans mostly as "neutral", but "cool" can also happen. In these cases, air temperature is the most important factor in the shaping of the thermal environment.

The same, but for the city of Szeged, is presented in Figure 5a,b.

The basic features determined for Sopron are also valid for Szeged. At noon, the thermal load in winter and summer in the majority of cases is 1–1.5 clo and −1−−2.5 clo, respectively. The effect of global radiation and wind speed on the evolution of thermal load around noon can also be observed. On the days (days 3, 4, 10, 14 and 15) with the largest thermal loads ($r_{cl}$ values around −2.5 clo) the incoming global radiation is 650–820 Wm$^{-2}$ and the wind is weak (1.5 m s$^{-1}$) or moderate (2.5 m s$^{-1}$). The related operative temperatures are more than two times larger than the air temperatures. On day 15 (26 June 2019), the operative temperature is 70.7 °C, the air temperature is 32 °C. The large $r_{cl}$ difference between days 11 (22 June 2019) and 12 is partly caused by the large global radiation difference (about 330 W m$^{-2}$). Besides global radiation difference, in this case the air temperature difference (about 8 °C) is also important. Similarly, in the early morning, the winter and summer thermal loads vary mostly between the ranges 1–1.7 clo and 0–0.7 clo, respectively. Regarding thermal sensation, the results obtained for Szeged are very similar to the results obtained for Sopron. Of course, in some cases there are also pronounced differences between the two cities, but they will be discussed in detail later in Section 5.4.

Regardless of which city is considered, both humans reacted in the same way in terms of thermal load to weather variations; the reactions differed only in the amount of thermal load, not in nature. The largest thermal load difference between humans 4 and 5 is about 0.25 clo; such differences are observed in both cities both at noon and in the morning. The differences in summer mornings are less, they never reach 0.25 clo. The systematic shift in the number of thermal loads between humans 4 and 5 is caused by the difference in total metabolic heat flux density. Since human 4 possesses a larger M than human 5 (Table 1), its $r_{cl}$ value in the same weather is always somewhat lower with respect to the $r_{cl}$ value of human 5. As we can see, an $r_{cl}$ difference of 0.2–0.3 clo is not so large as to make a difference in thermal sensation.

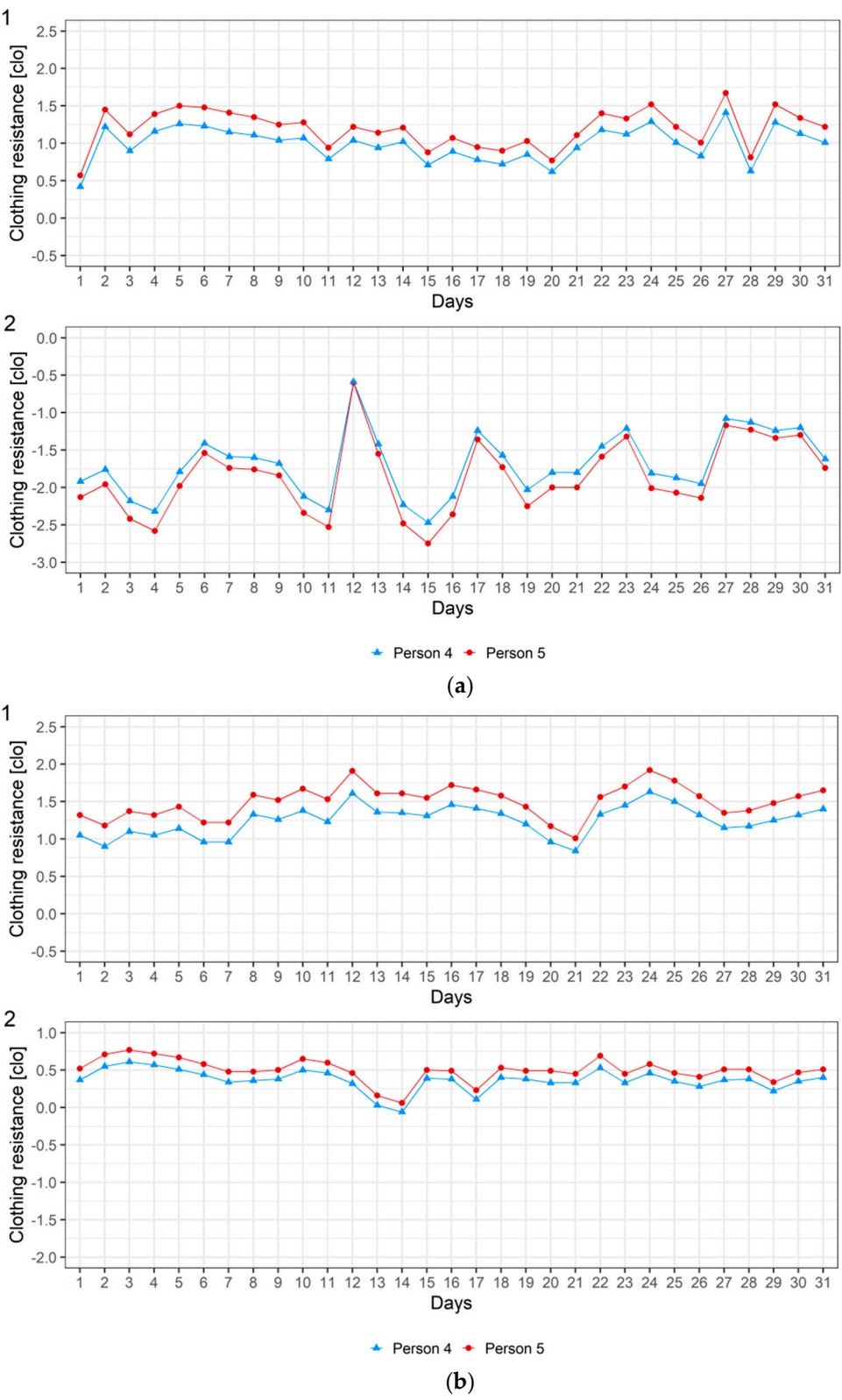

**Figure 5.** (**a**) Evolution of the thermal load at noon (12 UTC) for humans 4 and 5 over 31-day winter (**1**) and summer (**2**) season periods in the city of Szeged; (**b**) Evolution of the thermal load in the early morning (4–6 UTC) for humans 4 and 5 over 31-day winter (**1**) and summer (**2**) season periods in the city of Szeged.

### 5.4. Comparison of the Effect of Inter-Person Variation and the Effect of Weather Variation between the Cities of Sopron and Szeged

Inspecting Figures 4 and 5, it is hard to see differences in the thermal loads and thermal sensations between the cities of Sopron and Szeged for the same person. To be able to see these differences easily, they are presented separately in Figure 6 for human 5. Human 5 is chosen since in her case thermal load variations are larger than in the case of human 4.

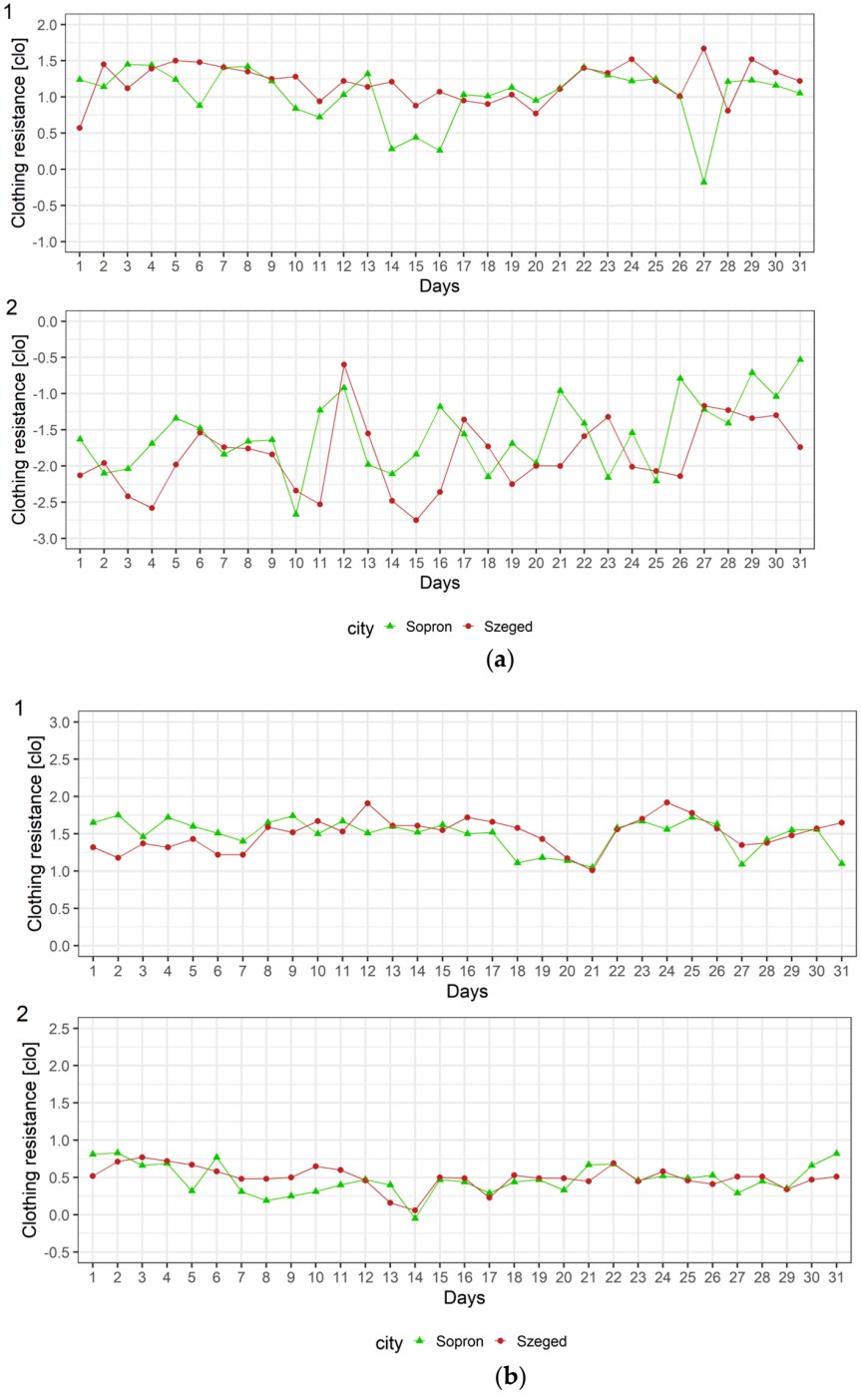

**Figure 6.** (**a**) Evolution of thermal load at noon (12 UTC) in the cities of Sopron and Szeged over 31-day winter (**1**) and summer (**2**) season periods for human 5; (**b**). Evolution of thermal load in the morning (4–6 UTC) in the cities of Sopron and Szeged over 31-day winter (**1**) and summer (**2**) season periods for human 5.

Comparing Figures 5a and 6a, as well as Figures 5b and 6b, we can get an insight into how large the effect of inter-person variation is on thermal load with respect to the effect of weather variation between the cities of Sopron and Szeged. In the following, the former effect will be briefly called the inter-person variation effect, while the latter the site variation effect. The main findings can be summarized as follows: (1) in the vast majority of cases the inter-person variation effect is comparable to the size variation effect, (2) the two effects seem to be comparable in summer morning cases, however, (3) there are such cases as well when the site variation effect is much greater than the inter-person variation effect, (4) the number of such cases at noon in summer is nine (about 30 percent of cases), and at noon in winter six (about 20 percent of cases), (5) the largest site variation effects can be as large as 1–2 clo, for instance, the largest site variation effect of 1.9 clo is observed at noon in winter on day 27 (Figure 6(a1)). (6) At noon in the summer, the site variation effect is between 0.6–1.2 clo, (7) these pronounced differences are also remarkably reflected in terms of thermal sensation, so, for instance, it can change from "neutral" to "cold" in winter, or, from "neutral" to "warm" in summer.

## 6. Discussion

According to Ács et al. [41], the Köppen climate formula of the cities of Sopron and Szeged is Cfb (C, warm temperate; f—no seasonality in the annual course of precipitation; b—warm summer). According to Ács et al. [54], the Feddema climate type of the cities of Sopron and Szeged is "cool and dry with extreme variations of temperature". According to Ács et al. [41], the human bio-climate of the cities of Sopron and Szeged, characterized in terms of clothing resistance, can be slightly different and depends also upon human somatotype. The mean annual clothing resistance value for human 1 is 0.4–0.8 clo [40,41]; accordingly, the corresponding thermal sensation may be "cool" and/or "neutral". So far, the spatial and/or temporal variations of the human bio-climate are investigated exclusively in terms of PET [55,56]. To broaden the methodological tools and the situations investigated, thermal load in terms of clothing resistance and the corresponding thermal sensations are estimated in different weather conditions. Weather variations in the cities of Sopron and Szeged are analyzed over 31-day winter and summer season periods at noon and early in the morning. The M of the considered humans 4 and 5 is 128 and 145 $\text{Wm}^{-2}$, respectively. Note that M of an "averaged" Hungarian male and female is 147 and 135 $\text{Wm}^{-2}$; that is, they practically fall into the $\Delta$M range (M = $\text{M}^4$–$\text{M}^5$) of humans 4 and 5.

The numerical simulations performed (e.g., Table 4) reveal the importance of global radiation and wind speed in the formation of clothing resistance at noon in winter and summer periods. At noon in summer, the global radiation of 600–800 $\text{Wm}^{-2}$ in windless situations increases the heat load substantially, thereby increasing operative temperature up to 70–80 °C and decreasing $r_{cl}$ down to −3 clo. This radiation forcing is so strong that $r_{cl}$ values in shady (about −1.0 clo) and sunny (about −2.5 clo) positions can differ by about 1.5 clo or even more if the wind is the same (Table 4). Consequently, thermal sensation is also different; in a shady location it can be "neutral" or "slightly warm", while in a sunny location "warm" or "very warm" depending on individual human characteristics (Table 4). The importance of radiation forcing at noon in winter is also observable (day 27 in the case of Sopron). Of course, in this case the radiation forcing is lower (about 400 $\text{Wm}^{-2}$ at its maximum value), but in close to windless situations this is enough to decrease the $r_{cl}$ value by about 1 clo, increasing operative temperature by about 16 °C. The related thermal sensation will change from "cool" to "neutral" for the people studied. Kántor and Unger [57] also recognized the determinant role of radiation in the shaping of the human bioclimate. The role of exposure to sunlight is also studied in the work of Kántor [58]. However, the approach is completely different. The author investigated whether the exposure to sunlight in different seasons influences the subjective estimation of thermal state, especially the sensation of "neutral" state. Environmental thermal load was estimated using PET and the population surveyed was several thousands.

In addition to radiation, the effect of wind on human thermal load is also appreciable. This topic is investigated more, for instance, in the work of Ács et al. [40]. According to Ács et al. [40], the $r_{cl}$ values are larger for wind gusts than for average wind for about 0.05–0.10 clo, the corresponding $T_o$ value changes are 0.3–0.5 °C. The $r_{cl}$ deviations can reach 0.5–1 clo when there is heat stress and the wind is strong. This behavior is also confirmed by other bioclimatic index simulations. This can be seen, for instance, in the work of Charalampopoulos and Nouri [59].

In summary, the determinant role of global radiation and wind speed in the formation of heat load at noon in both summer and winter seasons is indisputable. However, in general, these two factors are small in the morning. In the morning, the determinant factor is air temperature. When the air temperature is around 0 °C, thermal load is around 1.5 clo, inducing the thermal sensation of "cold" or less frequently "cool" depending on human characteristics and mental state. Similarly, when the air temperature is around 20 °C, the thermal load is around 0.5 clo, which can be sensed either as "neutral" or as "cool". There are also situations when all three factors are working. This can be observed, for instance, on day 27 at noon in winter (Figure 6a) comparing the weather of Sopron and Szeged. An air temperature difference of 5 °C (Szeged is colder), a global radiation difference of 228 Wm$^{-2}$ (irradiation in Szeged is smaller) and a wind speed difference of 3.9 ms$^{-1}$ (Szeged is more windy) together caused an $r_{cl}$ difference of 1.65 and 1.85 clo for human 4 and 5, respectively. We know that these three factors contradict each other in depressions and anticyclones. In depressions, the temperature is cool, irradiation is low (cloudy weather) and it is windy, whilst in anticyclones temperatures are warm (summer) or cold (winter), irradiation is high, and wind is gentle. It is obvious that extreme heat ($r_{cl}$ values less than $-2$ clo) and cold ($r_{cl}$ values larger than 2 clo) stresses in the Hungarian lowland can appear in the period of the impact of an anticyclone.

The model used is physically well established, but it is not simple enough. The model can be simplified by simplifying the calculation of operative temperature. It is essential to do this, since the model will not be user friendly in everyday applications. Carrying out this simplification is a task for the future. The model's huge advantage is that it can be run equally with weather or climate data, so it can also be used for climate classification purposes [43].

## 7. Conclusions

Individual human thermal load and sensation estimates are made for different humans in different weather conditions. Firstly, individual thermal load–thermal sensation relationships are constructed; secondly, weather in the cities of Sopron and Szeged in different seasons (winter, summer) and times of day (morning and noon) is characterized from the point of view of human thermal load and sensation. Thermal load is simulated in terms of clothing resistance and operative temperature. Humans (two males and four females) are walking at a speed of 1.1 ms$^{-1}$ in outdoor conditions without sweating. Their body shapes are also analyzed by applying the Heath–Carter somatotype classification method.

From the results obtained, the following main conclusions can be drawn. (1) Total metabolic flux density deviation between the persons considered is less than 20 Wm$^{-2}$, which is easily noticeable in terms of body shape. (2) In the early summer mornings weather is sensed either as "neutral" or as "cool", in these cases the inter-person variation effect is small and negligible. (3) At noon in summer, the weather involving a large thermal load is sensed either as "warm" or as "very warm", in these cases the inter-person variation effect is clearly visible and cannot be neglected. (4) The scheme presented is suitable for quantifying the intensity of individually perceived heat and cold stresses in the Hungarian lowland. It is constructed for individual use and it can be applied for use with weather or even climate data. In summary, there are such weather events in the Hungarian lowland, especially in summer, in which the inter-person variation effect cannot be neglected, even if the total metabolic heat flux density differences are smaller than 20 Wm$^{-2}$.

**Supplementary Materials:** Atmosphere_2020_Supplement containing weather, thermal sensation, https://www.mdpi.com/2073-4433/12/1/84/s1, operative temperature and clothing resistance data collected by the Hungarian authors of this study. Table containing radiation data ($Q_0$ and $\alpha$) for each month and hour can also be found in the same file.

**Author Contributions:** Conceptualization, F.Á.; methodology, F.Á., A.Z.; software, F.Á., A.Z.; validation, F.Á., E.K., A.Z., B.K., Z.S. and L.A.M.V.; formal analysis, F.Á., E.K., A.Z., B.K., Z.S. and L.A.M.V.; investigation, F.Á., E.K., A.Z. and L.A.M.V.; resources, E.K. and A.Z.; data curation, F.Á., E.K. and A.Z.; writing—original draft preparation, F.Á. and A.Z.; writing—review and editing, F.Á. and E.K.; visualization, E.K.; supervision, F.Á., E.K. and A.Z.; project administration, A.Z.; funding acquisition, A.Z. All authors have read and agreed to the published version of the manuscript.

**Funding:** E.K. was supported by the Széchenyi 2020 programme, the European Regional Development Fund and the Hungarian Government (GINOP-2.3.2-15-2016-00028). The online publication costs have been funded by the Eötvös Loránd University.

**Institutional Review Board Statement:** The study was conducted according to the guidelines of the Declaration of Helsinki, and approved by the Institutional Review Board (or Ethics Committee) of Institute of Geography and Earth Sciences, University Eötvös Loránd.

**Informed Consent Statement:** Informed consent was obtained from all subjects involved in the study.

**Data Availability Statement:** Data is contained within the article or supplementary material.

**Conflicts of Interest:** The authors declare no conflict of interest.

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
