# Peer review of "Weather in the Hungarian Lowland from the Point of View of Humans"

_atmosphere, doi:10.3390/atmos12010084_

Round 1

Reviewer 1 Report

In this manuscript, the authors report the human thermal load and sensation based on a survey made in two cities of Hungary. I admire the works the authors have been done because this kind of survey is not easy to be conducted. This manuscript is well organized and the reported data is valuable as a reference for related researches. There is no serious problem with this manuscript. My specific comments are listed below.

Major comments:

  1. It can be noticed that the recorded human clothing and sensations have been carefully discussed. There is no problem with the applied methodology, however, other details of the recorded data (e.g. temperature, humidity, wind speed, etc...) might be useful for other researchers to generate other sensation parameters. So I suggest the author also open the data set in the appendix.
  2. The first paragraph in the discussion is too lengthy and is not easy to catch the points. It should be split into several paragraphs or be itemized. In each paragraph or item, please indicates the highlights. For example, the point of L345-L355 seems to highlight the temporal and spatial difference; L355-L364 is about radiation; etc...
  3. I would like to see discussions about the relationship between wind speed (and humidity) and the recorded clothing resistance. That is not clear in the current manuscript.
  4. Is the equation for calculating clothing resistance (rcl) appropriate (eq. 1 and eq. 2) basing on your result? 

Minor:

  1. L226-L228: "The rcl–To relationship is like a band, the bandwidth is about 0.5 clo, which is caused by M varying between 127 and 145 Wm-2". Why the bandwidth is caused by the variation of M, please make the statement clear.
  2. L359: Missing the reference for "the numerical simulation".
  3. Figure 4, 5, 6: It's good to report the time series of recorded clo. However, the authors should make additional figures to visualize the highlight points.
  4. L345, L346, & L348: I suggest to rewrite the sentences. A statement simply starts from a citation is good to read.

Author Response

Comments and Suggestions for Authors

In this manuscript, the authors report the human thermal load and sensation based on a survey made in two cities of Hungary. I admire the works the authors have been done because this kind of survey is not easy to be conducted. This manuscript is well organized and the reported data is valuable as a reference for related researches. There is no serious problem with this manuscript. My specific comments are listed below.

Major comments:

  1. It can be noticed that the recorded human clothing and sensations have been carefully discussed. There is no problem with the applied methodology, however, other details of the recorded data (e.g. temperature, humidity, wind speed, etc...) might be useful for other researchers to generate other sensation parameters. So I suggest the author also open the data set in the appendix.

ANSWER: THANK YOU VERY MUCH FOR YOUR SUGGESTION. PERSONAL OBSERVATIONS (METEOROLOGICAL AND THERMAL SENSATION DATA) WILL BE PROVIDED SEPARATELY FOR EACH PERSON IN A SEPARATE DOCUMENT AS SUPPLEMENTARY MATERIAL.

  1. The first paragraph in the discussion is too lengthy and is not easy to catch the points. It should be split into several paragraphs or be itemized. In each paragraph or item, please indicates the highlights. For example, the point of L345-L355 seems to highlight the temporal and spatial difference; L355-L364 is about radiation; etc...

ANSWER: THE PARAGRAPH MENTIONED IS SUBDIVIDED INTO FOUR PARTS. PARAGRAPH 1 BRIEFLY PRESENTS “THE HISTORY OF BIOCLIMATOLOGICAL TREATMENTS AND THE CURRENT STATUS” related to THE cities of Sopron and Szeged; PARAGRAPH 2 HIGHLIGHTS THE IMPACT OF RADIATION, PARAGRAPH 3 HIGHLIGHTS THE ROLE OF WIND AND PARAGRAPH 4 CONNECTS EXTREME THERMAL EVENTS TO ANTICYCLONES. PARAGRAPH 3 IS COMPLETELY NEW, PARAGRAPHS 1 AND 2 ARE SLIGHTLY MODIFIED.

  1. I would like to see discussions about the relationship between wind speed (and humidity) and the recorded clothing resistance. That is not clear in the current manuscript.

ANSWER: THIS IS DISCUSSED AT TWO PLACES. PLACE 1: IN SECTION 5.3, LINES 306-326, DISCUSSING TABLE 4. TABLE 4 IS A NEW TABLE INTRODUCED AT THE REQUEST OF THE REVIEWERS. PLACE 2: PARAGRAPH 3 IN SECTION 6. REMARKS REGARDING rcl SENSITIVITY TO WIND SPEED CHANGES.

  1. Is the equation for calculating clothing resistance (rcl) appropriate (eq. 1 and eq. 2) basing on your result? 

ANSWER: YES, IT IS APPROPRIATE, IT HAS BEEN CHECKED SEVERAL TIMES AND THE NUMERICAL VALUES ARE CORRECT.

Minor:

  1. L226-L228: "The rcl–To relationship is like a band, the bandwidth is about 0.5 clo, which is caused by M varying between 127 and 145 Wm-2". Why the bandwidth is caused by the variation of M, please make the statement clear.

ANSWER: THE EXPLANATION IS ON LINES 258-261. IT IS AS FOLLOWS:  

The formation of this band is caused by the vertical and horizontal spread of the points. The vertical spread of the points is caused by variations of M, whilst the horizontal spread by variations of weather. The vertical bandwidth is about 0.5 clo, which is caused by the variations of M (between 127 and 145 Wm-2) of the five persons.

  1. L359: Missing the reference for "the numerical simulation".

ANSWER: THE CLARIFICATION IS ON LINE 408. IT IS AS FOLLOWS: The numerical simulations performed (Table 4) reveal …

  1. Figure 4, 5, 6: It's good to report the time series of recorded clo. However, the authors should make additional figures to visualize the highlight points.

ANSWER: THE ROLE OF RADIATION AND WIND IN FORMATION OF rcl IS ILLUSTRATED BY THE DATA IN TABLE 4 (SECTION 5.3, LINES 309-323) AND IS DISCUSSED SEPARATELY BELOW. ADDITIONAL FIGURES HAVE NOT BEEN ADDED, WE TRIED TO FULFILL THE REQUEST WITH TABLE 4.

  1. L345, L346, & L348: I suggest to rewrite the sentences. A statement simply starts from a citation is good to read.

ANSWER: THE CLARIFICATION HAS BEEN MADE AT ALL THREE PLACES AND THE SENTENCES ARE CLEAR.

Submission Date

08 November 2020

Date of this review

24 Nov 2020 03:52:37

Az űrlap alja

© 1996-2020 MDPI (Basel, Switzerland) unless otherwise stated

Reviewer 2 Report

Review for manuscript 1012252

The manuscript with the title “Human thermal load and sensation of weather in the Hungarian lowland” deals with a method for the characterization of the climate due to human thermal load and sensation.

The approach has novelty, and the topic is interesting and of course it is suitable for the Atmosphere journal.

I can see that there is a new approach of climatic classification (if I understand it correctly), but I need some more clear statements about what the meaning and the potential usage of this is.

Nevertheless, the manuscript needs major revision to become publishable. The following comments are separated in two sections, the general and the specific in order to help authors on their revision process.

If the authors choose to proceed with the suggested revisions, in the next round we will focus deeper to the context and meaning of the research.

General comments

  • The title is not so informative. I suggest a straight title such as “ The climate and weather characterization due to human thermal sensation…”, if I understand the meaning of the manuscript correctly.
  • The abstract needs restructuring and enrichment. First, it needs the problem’s statement. What is the problem we face, without the proposed paper? What is the solution that suggest this research?
  • The introduction is noticeably short without a state-of-the-art part. Authors should enrich their introduction with review or comparative papers such as doi: 10.1016/j.scitotenv.2018.02.276, doi:10.3390/atmos10100580, doi: 1016/j.scitotenv.2020.140092.
  • Why is it beneficial to characterize something due to a more subjective method, with much more parameters and lot of uncertainty?
  • How do we know that there are no other factors that affecting the thermal sensation (e.g. food, drugs, thyroid functionality, indoor (home) temperatures)?
  • Is the 6 persons sample an adequate number to conduct a research of this kind? How do you know that the results are not an outcome of the physiology of those 6 persons?
  • The authors should take care of the way they input the citation in the text. For example, in ll. 348 it is more readable to write “According to Ács et. al [19] “ than “According to [19]. I strongly suggest altering all the relative citations anywhere it is applicable.
  • The references are few for such a research. They must enrich their manuscript with more high-quality documentation.

Specific Comments

ll.23 – 25: The sentence is unclear. Rewrite it

ll.64: Since each paper is self-explanatory, the authors must analysis the basic concepts and its major parts.

ll.209 The symbols of figure 3a and 3b about B.K. and Z.S look identical. The figure needs revision in order to become clear.

  1.  Figures 4a to 6b is probably not the correct choice. Authors must think something more professional and comfort for the readers eyes. The same 6 repetitive figures are boring and disturbing.

Author Response

Comments and Suggestions for Authors

Review for manuscript 1012252

The manuscript with the title “Human thermal load and sensation of weather in the Hungarian lowland” deals with a method for the characterization of the climate due to human thermal load and sensation.

The approach has novelty, and the topic is interesting and of course it is suitable for the Atmosphere journal.

I can see that there is a new approach of climatic classification (if I understand it correctly), but I need some more clear statements about what the meaning and the potential usage of this is.

ANSWER: THE MODEL CAN BE EQUALLY RUN ON WEATHER AND CLIMATE DATA YIELDING INDIVIDUAL THERMAL LOAD AND THERMAL SENSATION INFORMATION. OTHERWISE, IT HAS ALREADY BEEN APPLIED ON CLIMATE DATA IN THE CARPATHIAN REGION. AS IT IS HIGHLIGHTED, IT IS CONSTRUCTED FOR INDIVIDUAL USE, ACCORDINGLY IT USES PERSONAL HUMAN DATA, AT THE SAME TIME ITS HUMAN SUBUNIT IS EXTREMELY SIMPLE. WE BELIEVE IN THE PRINCIPLE OF ‘INDIVIDUALIZATION’, SO, WE DON’T FOLLOW THE APPROACH OF ‘STANDARDIZED HUMAN’. IN THE WORK, THIS INFORMATION CAN BE FOUND IN THE ABSTRACT (LAST SENTENCE), IN THE LAST PARAGRAPH OF SECTION 6 AND IN SECTION 7 UNDER POINT 4.   

Nevertheless, the manuscript needs major revision to become publishable. The following comments are separated in two sections, the general and the specific in order to help authors on their revision process.

If the authors choose to proceed with the suggested revisions, in the next round we will focus deeper to the context and meaning of the research.

General comments

  • The title is not so informative. I suggest a straight title such as “ The climate and weather characterization due to human thermal sensation…”, if I understand the meaning of the manuscript correctly.

ANSWER: WE CHANGED THE TITLE, THE NEW TITLE IS: WEATHER IN THE HUNGARIAN LOWLAND FROM THE POINT OF VIEW OF HUMANS. WE THINK, THIS TITLE REFLECTS THE POINT OF THE STUDY. THE WORD ‘WEATHER’ HAS BEEN ADDED TO THE KEYWORDS.

  • The abstract needs restructuring and enrichment. First, it needs the problem’s statement. What is the problem we face, without the proposed paper? What is the solution that suggest this research?

ANSWER: WE CHANGED THE ABSTRACT ACCORDING TO THE SUGGESTIONS. THE TOPIC IS CLEARLY DESCRIBED AND THE MAIN RESULTS OBTAINED ARE LISTED. IT IS ALSO OBVIOUS THAT THE STUDY POSSESSES CLIMATOLOGICAL ASPECTS AND APPLICATION POSSIBILITIES.

  • The introduction is noticeably short without a state-of-the-art part. Authors should enrich their introduction with review or comparative papers such as doi: 10.1016/j.scitotenv.2018.02.276, doi:10.3390/atmos10100580, doi: 1016/j.scitotenv.2020.140092.

ANSWER: THE INTRODUCTION HAS BEEN EXPANDED SUBSTANTIALLY. ABOUT 20 NEW CITATIONS HAVE BEEN ADDED REFERRING TO BOTH ‘INTERNATIONAL’ AND ‘REGION-SPECIFIC’ ASPECTS.

  • Why is it beneficial to characterize something due to a more subjective method, with much more parameters and lot of uncertainty?

ANSWER: THE HUMAN SUBUNIT OF THE MODEL WORKS WITH FOUR HUMAN STATE VARIABLES: BODY MASS (THE MOST IMPORTANT), BODY LENGTH, SEX AND AGE AND THIS IS COMPLETELY SUFFICIENT TO CHARACTERIZE PERSONAL METABOLIC ENERGY FLUX DENSITY (M). THE MODEL DOES NOT TREAT ANY OTHER THERMOPHYSIOLOGICAL PROCESS, THAT IS, IT CANNOT BE SIMPLER AS IT IS. THIS IS NEEDED FOR USER FRIENDLY APPLICATIONS. IN OUR CASE ‘SUBJECTIVE’ MEANS PERSONAL M  AND ‘MUCH MORE PARAMETERS’ MEANS FOUR HUMAN STATE VARIABLES.  ‘LOTS OF UNCERTAINTY’? IN OUR OPINION THERE IS NOT A LOT OF UNCERTAINTY, WE HAVE TO COMPARE THE SIMULATION RESULTS RELATED TO M WITH THE MEASUREMENT RESULTS RELATED TO M.  SUCH A COMPARISON IS A TASK FOR THE FUTURE. AS WE MENTIONED, WE BELIEVE THAT THE PRINCIPLE OF ‘INDIVIDUALIZATION’ IS THE PRINCIPLE TO BE FOLLOWED IN THE FUTURE.   

  • How do we know that there are no other factors that affecting the thermal sensation (e.g. food, drugs, thyroid functionality, indoor (home) temperatures)?

ANSWER: WE KNOW THAT THERE ARE OTHER FACTORS AND WE TRIED TO EXLUDED THEM AS MUCH AS POSSIBLE DURING OUR INDIVIDUAL OBSERVATIONS.

  • Is the 6 persons sample an adequate number to conduct a research of this kind? How do you know that the results are not an outcome of the physiology of those 6 persons?

ANSWER: THE 6 PERSONS DO NOT REPRESENT ANY POPULATION AT ALL. OUR GOAL WAS NOT TO REPRESENT ANY ‘POPULATION’ RATHER THE POSSIBLE VARIABILITY OF M  BETWEEN DIFFERENT PERSONS. AND WE WERE LUCKY BECAUSE THE HYPOTHESIZED VARIABILITY IS CAUGHT TO SOME EXTENT BY THESE 6 PERSONS. AS WE SAID THE ‘PHYSIOLOGY’ IN OUR MODEL IS REPRESENTED ONLY VIA M AND M IS A FUNCTION ONLY OF FOUR HUMAN STATE VARIABLES OF WHICH BODY MASS IS THE MOST IMPORTANT.   

  • The authors should take care of the way they input the citation in the text. For example, in ll. 348 it is more readable to write “According to Ács et. al [19] “ than “According to [19]. I strongly suggest altering all the relative citations anywhere it is applicable.

ANSWER: THANK YOU VERY MUCH FOR THIS REMARK, THE SUGGESTED CORRECTION HAS BEEN MADE AT ALL THE RIGHT PLACES.

  • The references are few for such a research. They must enrich their manuscript with more high-quality documentation.

ANSWER: WE TRIED TO FULLFIL THIS REQUIREMENT AS MUCH AS POSSIBLE. IN THE END WE OBTAINED ALMOST DOUBLE THE NUMBER OF CITATIONS THAN IN THE UNCORRECTED MANUSCRIPT.

Specific Comments

ll.23 – 25: The sentence is unclear. Rewrite it

ANSWER: THE SENTENCE HAS BEEN CORRECTED. THE CORRECTION IS IN LINES 25-27.  

ll.64: Since each paper is self-explanatory, the authors must analysis the basic concepts and its major parts.

ANSWER: THE MAIN SUPPOSITIONS USED IN THE MODEL ARE PRESENTED. THE STRUCTURE OF THE MODEL, AND THE CONCEPT OF THE CHARACTERIZATION OF THE PROCESSES ARE ALSO BRIEFLY DESCRIBED. THE RELATED CITATIONS ARE ALSO GIVEN. 

ll.209 The symbols of figure 3a and 3b about B.K. and Z.S look identical. The figure needs revision in order to become clear.

ANSWER: WE TRIED TO IMPROVE THE FIGURES TAKING INTO ACCOUNT THE REMARKS.

  1.  Figures 4a to 6b is probably not the correct choice. Authors must think something more professional and comfort for the readers eyes. The same 6 repetitive figures are boring and disturbing.

ANSWER: WE TRIED TO IMPROVE THE FIGURES TAKING INTO ACCOUNT THE REMARKS.

Submission Date

08 November 2020

Date of this review

17 Nov 2020 16:34:46

Az űrlap alja

© 1996-2020 MDPI (Basel, Switzerland) unless otherwise stated

SUMMARIZING: THANK YOU VERY MUCH FOR THE REMARKS AND SUGGESTIONS. WE WANT TO SAY THAT THE PART OF MODEL DESCRIPTION HAS BEEN EXPANDED BY EQUATIONS FOR CALCULATING GLOBAL RADIATION AND THE TERM RADIATIVE-CONVECTIVE RESISTANCE. THIS EXTENSION IS NEEDED IF THE USER WISHES TO USE THE DATABASES GIVEN IN THE SUPPLEMENTARY MATERIAL.

Budapest, 13 December 2020

Round 2

Reviewer 2 Report

The authors made all the appropriate revision in their manuscript. Also, they answered clearly to my comments and suggestions. So, the manuscript could go further to the publishing process. The only advice I have is to change the axis scales to avoid wide empty areas in their graphs

Author Response

Revision-2

of the manuscript atmosphere 1012252

All suggestions are accepted, consequently, the corrections are made according to the suggestions. The file with corrections is named atmosphere-1012252-decision.v2. The corrections can be found on line as follows:

WU1 – line 47,

WU2 -  lines 50, 51,

WU3, WU4 – line 114,

WU5 – line 209,

WU6, WU7 – lines 252, 253,

WU8 – line 260 and

WU9 – line 292.

30th December, 2020, Martonvásár, Hungary,    Ferenc Ács.
